# Inhomogeneity of Free Volumes in Metallic Glasses under Tension

**DOI:** 10.3390/ma12010098

**Published:** 2018-12-29

**Authors:** Wei Da, Peng-wei Wang, Yi-fu Wang, Ming-fei Li, Liang Yang

**Affiliations:** College of Materials Science and Technology, Nanjing University of Aeronautics and Astronautics, Nanjing 210016, China; wei.d@nuaa.edu.cn (W.D.); 15705182552@163.com (P.-w.W.); wangyifu@nuaa.edu.cn (Y.-f.W.); lyglmf95@163.com (M.-f.L.)

**Keywords:** metallic glasses, synchrotron radiation, molecular dynamics, free volume, deformation, microstructure

## Abstract

In this work, the deformation of Zr_2_Cu metallic glass (MG) under uniaxial tensile stress was investigated at the atomic level using a series of synchrotron radiation techniques combined with molecular dynamics simulation. A new approach to the quantitative detection of free volumes in MGs was designed and it was found that free volumes increase in the elastic stage, slowly expand in the yield stage, and finally reach saturation in the plastic stage. In addition, in different regions of the MG model, free volumes exhibited inhomogeneity under stress, in terms of size, density, and distribution. In particular, the expansion of free volumes in the center region was much more rapid than those in the other regions. It is interesting that the density of free volumes in the center region abnormally decreased with strain. It was revealed that the atomic-level stress between different regions may contribute to the inhomogeneity of free volumes under stress. In addition, the inhomogeneous change of free volumes during the deformation was confirmed by the evolution of local atomic shear strains in different regions. The present work provides in-depth insight into the deformation mechanisms of MGs.

## 1. Introduction

Metallic glasses (MGs) have many potential applications due to their unique physical, chemical, and mechanical properties, such as relatively high levels of strength and hardness [1,2,3], a superior large elastic limit [4,5], excellent corrosion resistance, high wear resistance [6]. In particular the unique mechanical properties of MGs have attracted intensive interest. However, so far, compared with crystalline alloys, there is still a challenge to reveal the deformation mechanisms of MGs due to their disordered nature [7,8,9]. To address this issue, various specific structural concepts or models have been proposed, including shear transformation zones (STZs), flow units, and flexible volumes [10,11,12,13,14]. It was proposed that deformation occurs preferentially in some local regions, referred to as STZs [11], which are characterized by a lower density of atomic packing [10]. The coalescence of these STZs can form a shear band [15]. However, it is difficult to experimentally measure the shear localization at the atomic scale due to the extremely short time and the very small scale lengths involved [16].

As a specific concept, free volume was proposed by Turnbull [17,18,19], and widely applied to explain the glass formation in alloys [20,21,22]. It is supposed that free volumes may significantly affect the deformation of MG. In particular, it was revealed that free volumes in a shear band increase with strain, thereby decreasing the density of MG material and its resistance to deformation [23]. In addition, it was suggested that shear viscosity can be appreciably reduced in locally dilated regions with high concentrations of free volumes [15], and that the plastic deformation is related to the increase of free volumes [15,24]. In general, free volume is regarded as a key factor affecting the deformation of MGs. The revealing of the deformation mechanisms from the free volume aspect was expected. So far, although a number of experimental and theoretical investigations have been devoted to studying the free volumes of MGs [25,26,27], it is still a challenge to quantitatively or directly detect or characterize the free volumes, such a concept is ambiguous and elusive. Therefore, to understand the deformation mechanisms in terms of free volume is still a long-standing issue [27].

In this work, a feasible scheme for quantitatively calculating free volumes was developed and the deformation mechanisms characterized by the free volumes of MG were studied. A Zr_2_Cu binary composition was selected as the research prototype for the following reasons: (1) Zr_2_Cu has both amorphous and crystalline phases [28], which made it easy to calculate free volumes, and (2) the simple system allowed us to study the microstructure of MGs under stress.

## 2. Experiments and Simulations Section

An alloy ingot with a Zr_2_Cu composition was fabricated by arc-melting Zr and Cu, with a purity of 99.9 wt.%, in a Ti-gettered high-purity argon atmosphere. The ingot was melted at least five times to ensure the compositional homogeneity. The corresponding amorphous ribbons were prepared by melt-spinning, producing a cross section of 0.04 × 2 mm^2^. Subsequently, a synchrotron, radiation-based, high-energy X-ray diffraction (XRD) measurement was performed to detect the fine structure features of this ribbon sample. This was performed at beam line BW5 in the HASYLAB in Germany. Furthermore, extended X-ray absorption fine structure (EXAFS) measurements for both Zr and Cu K-edge were carried out using transmission mode at beam line BL14W1 in the Shanghai Synchrotron Radiation Facility of China and beam line U7C in the National Synchrotron Radiation Laboratory of China. Additionally, all of the experimental synchrotron radiation data was normalized via a standard data-reduced procedure [29].

It is known that molecular dynamics (MD) simulation is a powerful tool for studying the structural information of MGs, but this method does not usually fit the experimental data [30,31,32,33,34,35,36]. On the other hand, reverse Monte-Carlo (RMC), simulating the synchrotron radiation experimental data, is another effective method for probing fine structural information in MGs, but it lacks the chemical potential for avoiding uncertainties during calculation [37]. To obtain a reliable structure model, the RMC and the MD method should be combined. In other words, based on the structural model obtained from RMC fitting with synchrotron-radiation XRD and EXAFS data, a further MD simulation was performed to modify this model. The MD simulation was performed using the large-scale atomic/molecular massively parallel simulator (LAMMPS) program [38]. This scheme is similar to that used for studying MGs in previous work [39].

The Zr_2_Cu structural model obtained by MD simulation upon the RMC result was a cube containing more than 40,000 atoms. Subsequently, this model was enlarged by reproducing itself along the X-direction five times so that we could study the stretching process in this enlarged model containing about 200,000 atoms. In order to solve the problem of the structural instability caused by the reproduction of these cubic models, this enlarged model was relaxed 100 ps at 300 K within the NPT (constant atom number, constant pressure, and constant temperature) ensemble under periodic boundary conditions, thereby providing a stable structure model. After that, in another MD simulation, this enlarged model was deformed by applying uniaxial tensile strain at a rate of 1 × 10^8^ s^−1^ along the X-direction. The temperature was maintained at 300 K. Periodic boundary conditions were imposed in both the Y- and Z-directions. As a result, the evolution of this structural model during deformation could be studied.

## 3. Results and Discussion

### 3.1. MD Simulation with Synchrotron Radiation Data

The structural information obtained from the MD with RMC simulated model of the Zr_2_Cu composition is shown in Figure 1, including the S(Q) curves, total pair distribution function G(r), and Cu/Zr K-edge EXAFS signals. Both the simulated S(Q) and the EXAFS curves fit well with the experimental curves, indicating that the MD simulation with the RMC result was successful. It was shown that, beside the first strong peak, there were no sharp peaks in the S(Q) curve, suggesting that this model was fully amorphous [37]. Based on this simulated structural model, atomic- and cluster-level structural information can be deduced. Because all atoms were “frozen” in the simulated structural model, the positions and sizes of atoms were determined, making it possible to probe the free volumes.

The simulated strain-stress curve of this enlarged Zr_2_Cu structural model under uniaxial tension along the X-direction is shown in Figure 2. This simulated curve is in agreement with those calculated in previous work [40]. It was found that the stress had a maximum value of 2.29 GPa at a strain of about 8%, indicating the yield strength. When the strain increased, the strength dropped immediately and reached a so-called quasi-steady stress flow [41] which was about 1.8 GPa. According to Cheng’s work [42], the difference between the yield strength and the quasi-steady stress flow related to the degree of softening during deformation; its magnitude may reflect the propensity for strain localization in the flow region.

To study the microstructural evolution during deformation, deformed structural models at strains of 2%, 4%, 8%, 12%, 16%, and 20%, were selected in this work. It is known that the deformation under an applied stress is accompanied by some short-range structural changes in the amorphous alloys [32,40,42,43,44]. Furthermore, because the as-prepared structural model is an enlarged one constructed by reproducing the original cubic model along the X-direction five times, and the uniaxial tension is also along the X-direction. This enlarged model was divided into three regions called the center, the transitional, and the marginal region, as shown in the inset of Figure 2.

### 3.2. Cluster Evolution under Stress

Because it was revealed that cluster-level structural change may relate to the deformation of MG [43], the evolution of various Voronoi clusters (VCs) was investigated using the Voronoi tessellation method [45]. Figure 3 shows the distribution of some major VCs in the structural model that were deformed at different strains. It was observed that the fraction of the <0,0,12,0> VC decreased with the increase of the uniaxial tensile strain. This was consistent with previous results [40,41,42,44,46]. It is known that <0,0,12,0> VC is the so-called full icosahedron that possesses abundant five-fold symmetries and is regarded as an indicator of high shear resistance, high packing density, and low potential energy [47]. During the process of deformation, it was proposed that both <0,0,12,0> and some distorted icosahedral VCs such as <0,1,10,2> [41], which are densely packed units, would collapse to form loosely packed ones. It was observed that the changes of the fractions of these major VCs with strains were very similar to each other in the three divided regions, suggesting that the cluster-level change was almost the same along the X-direction. In addition, these major VCs sharply decreased in all regions, indicating that all regions of this model were deformed by stress.

### 3.3. Atomic Structural Evolution under Stress

#### 3.3.1. Characterization of Free Volumes in MGs

It was suggested that, besides clusters, free volumes can also significantly contribute to the deformation in MGs [23]. In the present work, a new computational approach that can quantitatively characterize free volumes in amorphous alloys was designed. In one structural model, a probe sphere with a changeable radius was selected and was allowed to move randomly throughout the model. After it touched any of the position-determined atoms, a space that was not filled with atom(s) was detected. We called it unfilled space. In detail, the first step was to find the center position of this probe sphere in any tetrahedron made up of four neighboring atoms, by applying this equation:(1)$$D_{i}=D_{j},$$
where *D* was the distance between one neighboring atom and the center of a probe sphere, *i* and *j* denoted any two atoms of this tetrahedron. The volume of a probe sphere can be expressed by:(2)$$V_{PS}=\frac{4}{3}\pi R_{max}^{3},$$
where *R_max_* denoted the maximal radius of this sphere, and could be obtained by:(3)$$R_{max}=D_{i}-R_{i},$$
where *R_i_* was the radius of the *i*-th neighboring atom. The center of each unfilled space was strictly defined. Any two neighboring unfilled spaces whose center distances were too small or too large were merged or separated so that the overlapping of small unfilled spaces and the ignoring of large ones could be excluded. Figure 4 is a diagrammatic sketch for searching all possible unfilled spaces in a structural model. We emphasize that an unfilled space itself cannot denote the free volume directly [25]. These detected unfilled spaces were made up of both intrinsic unfilled spaces and free volumes [32].

In a crystalline alloy, the unfilled spaces are called intrinsic unfilled spaces. They are formed by the dense hard-sphere packing of atoms. Nevertheless, in MG, besides the intrinsic unfilled spaces, there are other unfilled spaces that are called free volumes. It is known that there is a competitive crystalline phase in the Zr_2_Cu composition [48], thus, it was suggested that the intrinsic unfilled spaces in the glassy state should be similar to those in this crystal phase [32]. All of the intrinsic unfilled spaces in the Zr_2_Cu tetragonal phase were calculated and the distributions are listed in Table 1. It was found that, in this crystal model, there was no intrinsic unfilled space with a radius larger than 0.4 Å, implying that intrinsic unfilled spaces in the corresponding Zr_2_Cu MG model should have radii shorter than 0.4 Å. In other words, any unfilled space with a radius larger than 0.4 Å should be a free volume rather than an intrinsic unfilled space. We have realized that some small free volumes with a radius less than 0.4 Å do occur in the structural model. By subtracting the small unfilled spaces in the crystal model from those in the MG model, some small free volumes could be estimated. However, such small free volumes only accounted for a small fraction, 4.5%, of the total free volume. In addition, with the increased strain, small unfilled spaces, including the intrinsic ones, and small free volumes decreased sharply, while large free volumes increased. Therefore, these small free volumes can be ignored. 

#### 3.3.2. Inhomogeneous Change of Free Volumes under Stress

The sizes of free volumes with the radius larger than 0.4 Å in the MG structural model under uniaxial tension were calculated and are shown in Figure 5a. It was observed that the average size of free volumes increased slowly with the increase of strain, from about 0.568 to 0.595 Å. Additionally, the sizes of free volumes in the different regions (center, transitional, and marginal) of this model were calculated. It was surprising that, although the sizes of free volumes in these regions also increased slowly with the increase of strain, the rates of rise between these three regions were different. This indicated that, unlike clusters, the size change of free volume is not the same in the different zones, implying an inhomogeneous nature of free volume in this MD model. The size differences between the as-prepared and the deformed models were calculated and displayed in Figure 5b. It is worth noting that, when strain reached 12%, the free volume size had an extraordinary increase of more than 3% in the center region, while that of the counterparts in the transitional and marginal regions was less than 1%. This implied that free volumes probably expand with the increase of strain throughout the enlarged model, in particular in the center region. We also noticed that the size increase of free volume apparently gets slower when strain is more than 8%, and remains constant when strain is more than 12%.

To learn more about the inhomogeneous expansion of free volumes during deformation, the total free volume in each region of this model was also calculated and is plotted in Figure 6. As expected, the evolution of total free volume had a similar trend to that of the size of free volumes. In particular, the total free volume in the center zone sharply increased from about 6.25 × 10^4^ to 6.85 × 10^4^ Å^3^ when strain was smaller than 8%. Subsequently, the total free volume slowly increased and reached an upper limit value of about 6.92 × 10^4^ Å^3^ when the strain was larger than 12%. The total free volume in both the transitional and the marginal zones was similarly dependent on the strain. However, the change of total free volumes in these two regions was much smaller than that of the center zone. This was in accordance with what is shown in Figure 5.

Unlike crystal alloys with well-defined structural units (the so-called unit cells) and well-defined defects (point defects or dislocations) that are sensitive to deformation, it has been proposed that the short-range order structures (VCs) and some “defects” (free volumes) affect the deformation of MGs. It has been suggested that some VCs, such as <0,0,12,0>, are sensitive to strain and apt at changing into relatively loosely packed clusters. Such transformation was probably stopped in the plastic stage (referring to Reference [30] and this work). We suppose that this was because of the relatively small/large energy and structural barriers of this cluster-level transformation at relatively small/large strains. Meanwhile, from both energetic and structural aspects, free volumes should have an upper bound when MG is stretched [49]. When the strain is small, it is easy for free volumes to increase with strain to achieve new energy balance and structural stability. After the 8% strain, it probably gets difficult to go on expanding free volumes, leading to an upper limit that indicates a saturation of free volumes [30].

Although the information of free volumes shown in Figure 4 and Figure 5 implies the inhomogeneous distribution of free volume during deformation, more direct evidence is required. Here, the number density of free volumes (*η_fv_*) was calculated via the following equation:(4)$$\eta_{fv}=\frac{N_{fv}}{V_{u}},$$
where *N_fv_* and *V_u_* are the number of free volumes and the space that contains these free volumes, respectively. The change of number density may directly reflect the inhomogeneous nature of free volumes during deformation. To explicitly demonstrate this evolution, a correlation between the number density of free volumes and strain is plotted in Figure 7. It is interesting that the density of free volumes decreased slowly with the increase of strain in the center region; this was quite different from other zones and abnormal because it has been suggested that more free volumes should appear when strain increases [50]. Nevertheless, we noticed that the number density decrease in the center zone was much smaller than the increase of free volume size, implying that total free volumes were probably more influenced by size than density. Therefore, it is possible that, in the center zone, the total free volume had an increasing trend similar to that of size rather than density. In addition, it is worth noting that the decrease or increase of free volume density in different zones also slowed when the strain was larger than 8% and remained constant when the strain reached 12%. This was consistent with the patterns shown in Figure 4 and Figure 5. We tried to fit data of the center, transitional, and marginal zones, as well as the average zone. We found that the best fitting lines of all these data were well described by the exponential function:(5)$$Y=Ae^{-BX}+C,$$
where *Y* and *X* denoted the density of free volume and the strain, respectively. Additionally, *A*, *B*, and *C* were coefficients or constants. All the values of *A*, *B*, and *C* of all the fitting lines are listed in Table 2. The exponential function explained why the free volumes were saturated when the strain was large. Moreover, it is interesting that the A value of the center region was positive while the counterpart of the transitional, marginal, or average was negative.

In addition, we know that bulk MGs with a thickness of more than 1 mm usually have good glass-forming ability and almost no plasticity under tension. However, it is acknowledged that a smaller MG may have better plasticity. In previous work studying the deformation behavior of the ZrNi thin film MGs [51,52], the plasticity apparently increased with the decrease of the thickness from 550 to 110 nanometers. For our model, it should be a very small MG because its size was only 38.0 nm × 14.5 nm × 7.2 nm, much smaller than that of the ZrNi thin film. Therefore, our model had a long strain range, indicating good plasticity. In this sense, the relatively large plasticity was due to size effect rather than uncertainties of MD simulation.

#### 3.3.3. Atomic-Level Stress between Different Zones

We supposed that the inhomogeneous distribution of free volumes under tension, in particular the abnormal density change of free volumes in the center zone, was probably related to atomic-level stress in this model. To examine this scenario, further analysis is required. The atomic percentages of the center, transitional, and marginal regions of the whole model, were calculated and are listed in Table 3. It was found that, unlike the other two zones, the atomic fractions of the center region got smaller with the increase of strain. In other words, a few atoms migrated from the center zone to other regions, implying the appearance of atomic-level stress between these zones, as illustrated in Figure 8. The combination of applied uniaxial stress and atomic-level stress made the atomic percentage or the density of free volumes in the center zone different from their counterparts in the other two regions. The center region was under larger total tensile stress, made up of the uniaxial tensile stress and the atomic-level tensile stress, which led to a rapid swelling of free volumes in the center zone. When the strain reached 12%, the atomic percentage became constant in all three regions, suggesting that atoms hardly left the center zone from then on, so that the atomic-level stress between these regions probably disappeared. As a result, the free volumes were saturated in all three regions. Since the scale of deformation-induced internal stresses was much bigger than the size of our computer model, this model could not contain the shear bands that can form in real MGs, accompanied by long-range (macroscopic) internal stresses [53,54]. Therefore, the deformation-induced internal stresses could not be discussed in this work.

We have revealed that there is an abnormal decrease in density of free volumes in the center zone. This had to be explained. It was revealed that the center zone is under a larger total tensile stress, a combination of uniaxial tensile stress and atomic-level tensile stress. This total tensile stress caused the rapid swelling of free volumes. This expansion may have increased the possibility to connect or merge separated free volumes [55], leading to the abnormal decrease in density of free volumes despite the increased total free volume. The evolution of free volumes in the center zone was studied. It was found that neighboring free volumes merged with each other when subject to increased strain, as shown in Figure 9. Concerning the transitional and marginal regions, the total tensile stress was smaller than that of the center zone at the same strain. The expansion of free volumes was also smaller, leading to a lower possibility of connecting or merging the neighboring free volumes. Therefore, unlike the center zone, the density of free volumes in the transitional and marginal zones increased rather than decreased with the strain.

#### 3.3.4. Evolution of Atomic Shear Strain under Stress

It has been suggested that the so-called local atomic shear strain, *η_i_^Mises^* [56], denotes the atomic motions in local regions under stress and is related to the density of shear transformations [57,58]. Meanwhile, the atomic motions probably affect the distribution of free volumes. Therefore, we wondered whether the change of atomic shear strain was similar to that of free volumes in different zones. Figure 10 shows a sequence of images that demonstrate the evolution of *η_i_^Mises^* in the center, transitional, and marginal regions. It was observed that, when the strain was about 2%, a number of atoms were activated by stress and atoms with relatively large *η_i_^Mises^* values were apt to separate into local regions or clusters, such as those marked with circles. These separated local regions were very similar to the local flow units of small strains [13]. When strain increased to 4% local regions got larger or more atomic-activated regions appeared. It is worth noting that, in the center zone, the local regions with the most active atoms propagated, coalesced, and nucleated, implying the formation of STZs, while STZs were still absent in the transitional and marginal zones. When the strain increased to about 8%, STZs formed by the propagation of atomic-activated local regions and appear in all three zones. Nevertheless, in the center zone, the *η_i_^Mises^* values in the STZs were apparently larger than those in the transitional or marginal zones. All of this indicated that atoms in the center zone were more likely to be activated by stress, concentrating in local regions where it was easier to form STZs. This is consistent with the inhomogeneous distribution of free volumes in the center, transitional, and marginal regions.

### 3.4. Potential Studies on Micro-Mechanisms of Deformation

As we know, flow units in MGs are extremely inhomogeneous at high stress or low temperatures [59]. Flow units do not significantly differ from the surrounding areas in structure, but they tend to have more free volumes that make them more active and susceptible to external energy. When MG is deformed, these flow units are activated, leading to local flow deformation [60]. With the increase of strain, localized flow units may transform into plastic flow units [61] or form shear bands [62,63,64], resulting in plastic deformation [65,66,67]. In addition, it has been suggested that a shear band is initiated at a region with concentrated stress or excess free volumes [23,68] and starts around the region of free-volume concentration with a scale of several atomic diameters [69,70]. During the deformation in MG, free volumes in a shear band are expected to increase, thereby decreasing the density of this material and its resistance to deformation. In this work, we revealed that, during a tensile deformation, there were some atomic-level changes, in terms of free volumes, atomic-level stress, and atomic shear strain. The evolution of free volumes was divided into three stages: rapid and slow expansions, and saturation, roughly corresponding to the elastic, yield, and plastic stages, respectively. In addition, free volumes exhibited an inhomogeneous response in different regions under tension. In the future, we expect to study the deformation of MGs by combining the flow unit, the STZ, and the free volume concepts or models.

## 4. Conclusions

In summary, a new computational method for quantitatively detecting free volumes was developed. The underlying deformation mechanisms of MGs were investigated from the free volume aspect. It was revealed that free volumes expanded with increased of strain and this expansion stopped in the plastic stage. In addition, although free volumes were homogeneously distributed throughout the as-prepared structural model, free volumes exhibited inhomogeneity in different zones of this model under tension in terms of their size, density, and distribution. In particular, the expansion of free volumes in the center region was much more rapid than in other places and the density of free volumes decreased abnormally in the center zone. The inhomogeneous change of free volumes under stress was due to the atomic-level stress between the different zones and could be validated by the evolution of local atomic shear strains in these zones. This work may shed light on the micro-mechanisms of deformation in glassy alloys.

## Figures and Tables

**Figure 1 materials-12-00098-f001:**
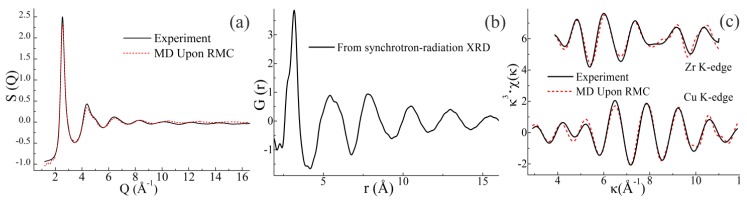
Synchrotron radiation data of a Zr_2_Cu sample, including: (**a**) structural factor S(Q), (**b**) pair distribution function G(r), and (**c**) normalized Zr and Cu K-edge EXAFS curves. The solid and dashed lines denote the experimental data and the MD simulation upon RMC result, respectively. The κ and the χ(κ) represent the photoelectron wave vector and the κ-space EXAFS signal, respectively.

**Figure 2 materials-12-00098-f002:**
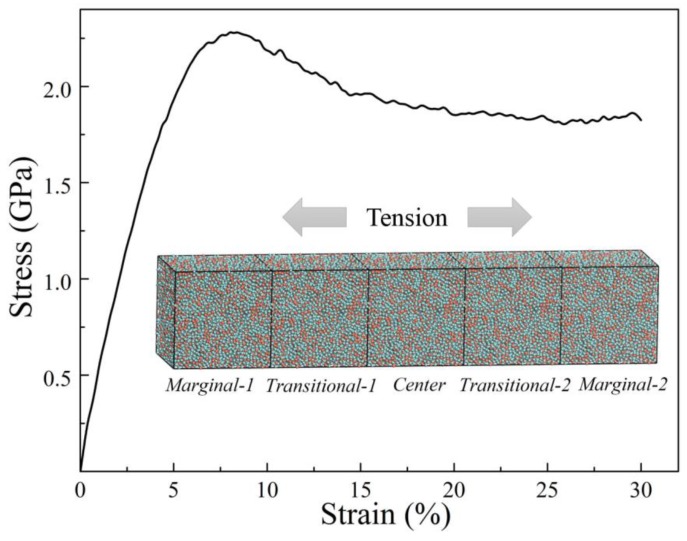
The MD simulated uniaxial stress-strain curve of the Zr_2_Cu structural model under tension. The inset is the enlarged model which was obtained by reproducing the MD and RMC simulated cubic model along the X-direction five times. This enlarged model was divided into three kinds of regions named center, transitional 1, 2, and marginal 1, 2.

**Figure 3 materials-12-00098-f003:**
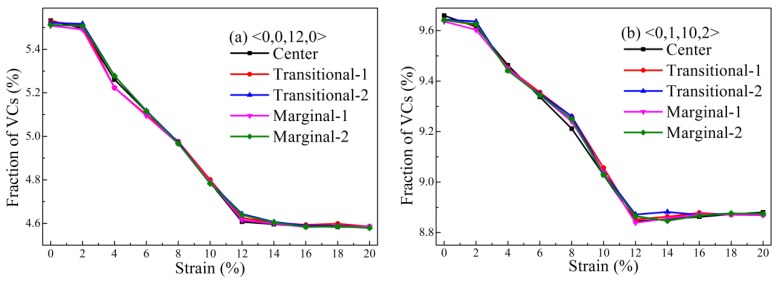
Distribution of some major VCs such as (**a**) <0,0,12,0> and (**b**) <0,1,10,2> in the center, transitional, and marginal regions of the structural model under tension.

**Figure 4 materials-12-00098-f004:**
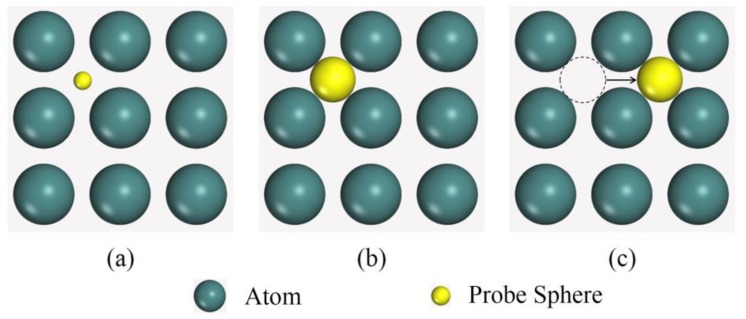
A diagrammatic sketch for searching all possible unfilled spaces in a structural model, including three steps: (**a**) location of an unfilled space, (**b**) size or volume evaluation of an unfilled space, and (**c**) the next unfilled space.

**Figure 5 materials-12-00098-f005:**
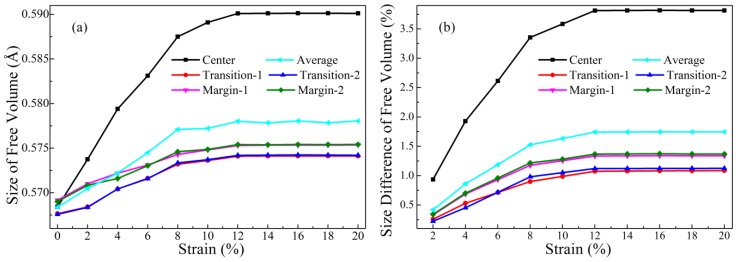
Relations between free volumes (large unfilled spaces whose radii are larger than 0.4 Å) and strain in tension mode, including: (**a**) the sizes of free volumes and (**b**) the difference ratio of sizes at different strains. All the curves of the center, transitional 1, 2, marginal 1, 2 regions and their averages are plotted.

**Figure 6 materials-12-00098-f006:**
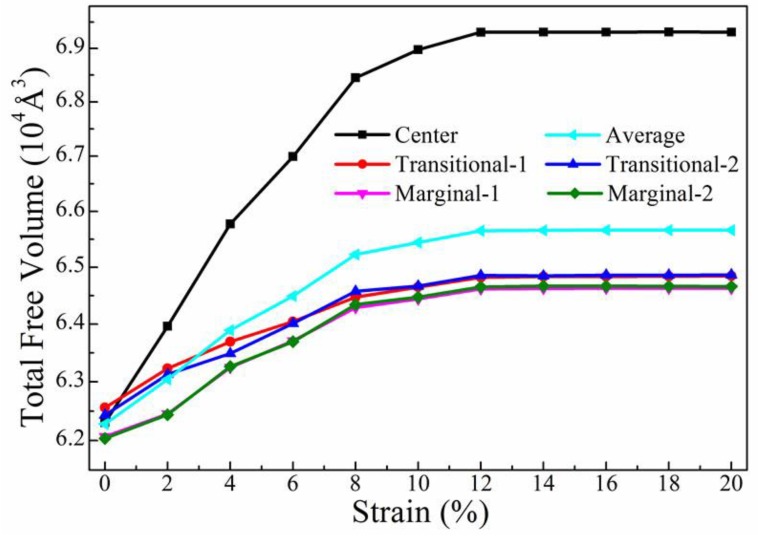
The total free volume in each region of this model under tension, as well as the average value.

**Figure 7 materials-12-00098-f007:**
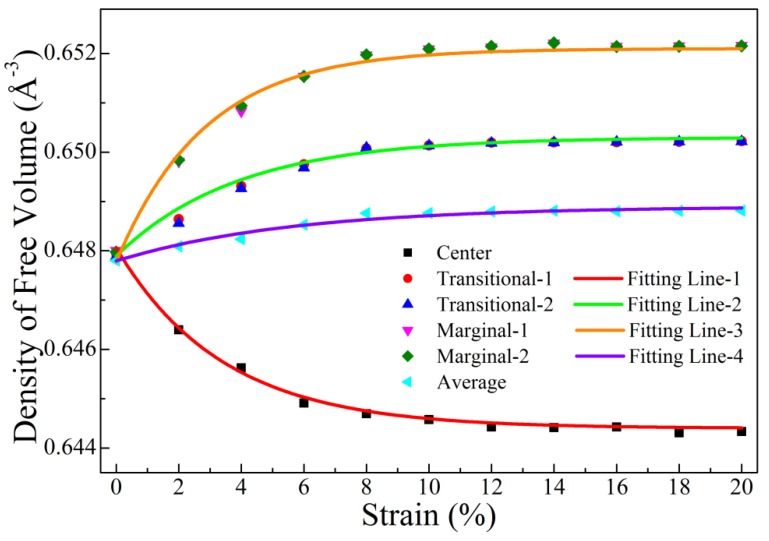
Number density of free volumes in each region of this model under tension, as well as the average value. The solid curves denote the fitting lines, which can be described by an exponential function.

**Figure 8 materials-12-00098-f008:**
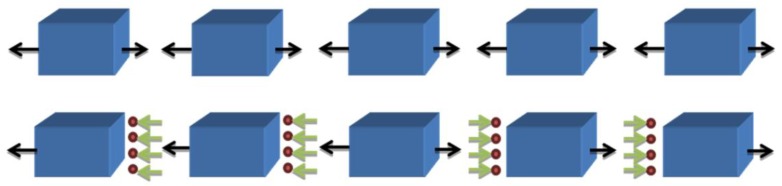
A diagrammatic sketch of the atomic-level stress between the three divided regions in this model. The upper and the lower panels indicate the stress corresponding to the 0% and 4% strains, respectively. The black and celadon arrows stand for the applied stress and atomic-level stress, respectively.

**Figure 9 materials-12-00098-f009:**
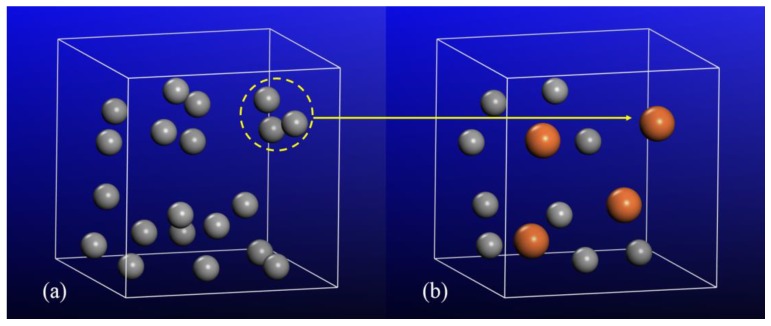
The distribution of free volumes in a local region (5 × 5 × 5 Å^3^) of the center zone at the strains of (**a**) 0% and (**b**) 4%. The gray and the red spheres denote free volumes with relatively small (0.4–0.8 Å) and large (>0.8 Å) sizes, respectively.

**Figure 10 materials-12-00098-f010:**
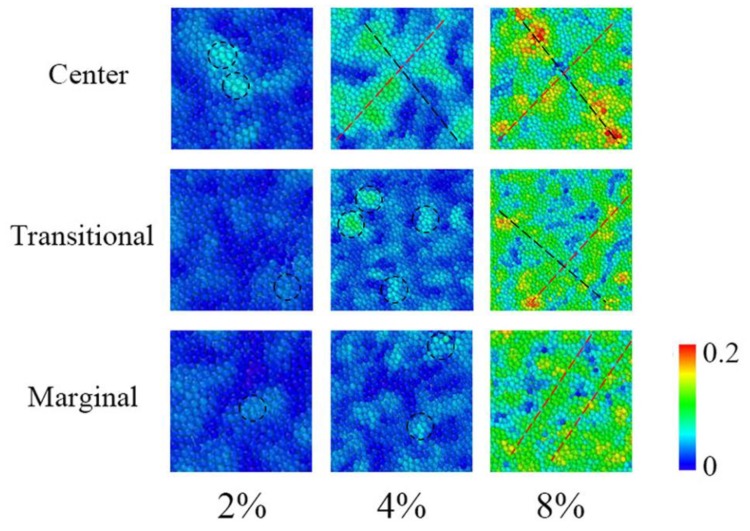
A sequence of images that demonstrate the evolution of *η_i_^Mises^* in the center, transitional, and marginal regions, corresponding to strains of 2%, 4%, and 8%, respectively. The circles indicate the localized regions with relatively high *η_i_^Mises^* values. The black and the red dashed lines denote STZs.

**Table 1 materials-12-00098-t001:** Size distribution of intrinsic unfilled spaces detected by a structural model containing more than 50,000 atoms of the Zr_2_Cu tetragonal phase.

	Percentage (%)			
Size (Å)	≤0.2	0.2–0.3	0.3–0.4	≥0.4
Zr_2_Cu	0	66.7	33.3	0

**Table 2 materials-12-00098-t002:** The values of *A*, *B*, and *C* of all the fitting lines. All of the fitting lines can be expressed by an exponential function. *A*, *B*, and *C* are coefficients or constants of this exponential function. Fitting lines 1, 2, 3, and 4 correspond to the center, transitional, and marginal regions, and the average one, respectively.

Coefficient	Fitting Line-1	Fitting Line-2	Fitting Line-3	Fitting Line-4
*A* (×10^−2^)	0.3659	−0.2407	−0.4322	−0.1106
*B*	0.2931	0.2577	0.3511	0.1772
*C*	0.6444	0.6503	0.6521	0.6489

**Table 3 materials-12-00098-t003:** Atomic percentages of the center, transitional, and marginal regions in the deformed structural models. Note that all of these values are nearly 20% when the strain is zero.

Strain	Atomic Percentage (±0.0005%)				
	Center	Transitional-1	Transitional-2	Marginal-1	Marginal-2
0%	20.0029	19.9967	19.9962	20.0012	20.0030
2%	19.9348	20.0106	20.0105	20.0216	20.0225
4%	19.8685	20.0211	20.0212	20.0435	20.0457
6%	19.8382	20.0278	20.0278	20.0529	20.0533
8%	19.8103	20.0334	20.0335	20.0611	20.0617
10%	19.8054	20.0341	20.0342	20.0631	20.0632
12%	19.7989	20.0351	20.0349	20.0655	20.0656
14%	19.7976	20.0353	20.0354	20.0656	20.0661
16%	19.7975	20.0357	20.0356	20.0655	20.0657
18%	19.7969	20.0358	20.0359	20.0656	20.0658
20%	19.7973	20.0358	20.0358	20.0655	20.0656

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
