# Peer review of "Inhomogeneity of Free Volumes in Metallic Glasses under Tension"

_materials, 2018, doi:10.3390/ma12010098_

Reviewer 1 Report

The authors provide a paper dealing with the inhomogeneity of free volumes in metallic glasses under tension. I found the paper of particular interest and quite well written focusing on key research topic i.e. the evolution of free volume for metallic glass deformed in tension. However, some revisions are needed:

1.       I would like that the authors comment about real experimental data and how this study can contribute in the understanding the deformation behavior of metallic glass. Specifically, in doi.org/10.1016/j.actamat.2017.03.072 the authors can see that ZrNi thin film metallic glass can exhibit a large plastic behavior while keeping an amorphous structure but probably with disruption of local order and free volume generation during tension. In addition, other papers report on the extrinsic origin of this size effects related to a thickness confinement doi.org/10.1016/j.actamat.2015.02.038. In this context, I would like that the authors discuss how their analysis can explain the mechanical properties of metallic glass in thin film form discussing the elastic plastic behavior and free volume generation in tensile test.

2.       Related to the previous point, I would like that the authors comment on validity of their MD simulations to explain the mechanical behavior of metallic glass (especially thin films). What are the number of atoms considered? Is that reliable for bulk and thin film metallic glasses? What are the assumptions made by the authors for the Zr-Cu bonds?

Author Response

Reviewer 1

The authors provide a paper dealing with the inhomogeneity of free volumes in metallic glasses under tension. I found the paper of particular interest and quite well written focusing on key research topic i.e. the evolution of free volume for metallic glass deformed in tension.

Reply: Thanks the referee very much for his/her positive comments.

1) I would like that the authors comment about real experimental data and how this study can contribute in the understanding the deformation behavior of metallic glass. Specifically, in doi.org/10.1016/j.actamat.2017.03.072 the authors can see that ZrNi thin film metallic glass can exhibit a large plastic behavior while keeping an amorphous structure but probably with disruption of local order and free volume generation during tension. In addition, other papers report on the extrinsic origin of this size effects related to a thickness confinement doi.org/10.1016/j.actamat.2015.02.038. In this context, I would like that the authors discuss how their analysis can explain the mechanical properties of metallic glass in thin film form discussing the elastic plastic behavior and free volume generation in tensile test.

Reply: I agree with the editor that we should discuss about the real experimental data and how this study can contribute in the understanding the deformation behavior of metallic glass (MG), and thank the reviewer very much for recommending these two articles. In brief, I strongly agree with the reviewer that the difference between simulation (such as our work) and real experimental data in particular the plastic behavior, is not caused by methodology itself, but is due to the “size effect”. As we know, bulk metallic glasses (BMG) with a thickness more than 1 mm usually have good glass-forming ability, while almost have no plasticity under tension. However, it is acknowledged that that a smaller MG may have a better plasticity. Especially, as suggested in these two articles studying the deformation behavior of the ZrNi thin film MGs, the plasticity apparently increases with the decrease of the thickness from 550 to 110 nanometers. For our model, it should be a very “small MG” because its size is only 38.0 nm X 14.5nm X 7.2nm, even much smaller than that of the ZrNi thin film. Therefore, our model has a long strain range, indicating good plasticity. In this sense, the relatively large plasticity is due to the size effect rather than uncertainties of MD simulation. I suppose that if a real sample as small as our model could be obtained, it probably has an excellent plasticity measured in real experiment. I will add the above analysis in the revision manuscript.

In addition, concerning the discussion of the elastic plastic behavior and free volume generation in tensile test, I think that the change of free volumes may relate to the elastic and yield stages, as discussed in our work and other articles. However, the overall change of free volumes, in terms of the weight, the distribution, the size, and so on, may not significantly affect the plasticity, because it seems that free volumes in our model get saturated when the softening of this model takes place from the strain about 10%, in accord with previous work that there is a critical fraction about 2.4% for reduced free volume from the onset of yielding in both brittle and plastic MGs (Wang, J. G.; Zhao, D. Q.; Pan, M. X.; Wang, W. H.; Song, S. X.; Nieh, T. G., Correlation between onset of yielding and free volume in metallic glasses. Scr. Mater. 2010, 62, 477-480). However, based on our work studying the “Inhomogeneity of free volumes in metallic glasses under tension”, in the future, we can further study the change of free volumes in some local regions especially in or close to the places where shear initiation, development, and transformation may occur. This may be helpful to reveal the structural origin of good plasticity in MGs, because it has been suggested that the free volume evolution in the localized shear zones may explain the brittleness or plasticity in MGs. Anyway, in this work, it is still difficult to discuss the plasticity in MGs, whereas we are expecting to address this issue.

2) Related to the previous point, I would like that the authors comment on validity of their MD simulations to explain the mechanical behavior of metallic glass (especially thin films). What are the number of atoms considered? Is that reliable for bulk and thin film metallic glasses? What are the assumptions made by the authors for the Zr-Cu bonds?

Reply: Although compared with the real experiment, there are some problems when we performing a tension test upon a MG model in MD simulation, such as the relatively fast strain rate, we can validate our MD simulation using free volume data to discuss the mechanical behavior in MGs, because: 1) since our model is smaller than the thin films, and even much smaller than BMGs, it is reasonable that our model can exhibit the “best plasticity”, the thin films have “good plasticity”, while BMGs usually have “poor plasticity”; 2) it is found that free volumes in our model get saturated from the strain about 10%, in accord with previous work that there is a critical fraction about 2.4% for reduced free volume from the onset of yielding in both brittle and plastic MGs. This indicates the reliability of analysis of free volumes in our model. Therefore, I suppose that the “Inhomogeneity of free volumes in metallic glasses under tension” may be an intrinsic feature in our model, thin films, or BMGs, although it is still difficult to perform the in-situ measurement of free volumes in thin films, or BMGs that are under stress.

There are about 200,000 atoms in our enlarged model to perform the MD simulation.

The assessed Zr-Cu bond in the as-prepared ZrCu MG model is about 2.85 Å, which could be deduced from the partial pair distribution of this model, as shown below. This value is close to the sum of Cu and Zr Goldschmidt atomic radius, and as same as that in previous work:

1. L. Yang, ect., Atomic structure in Al-doped multicomponent bulk metallic glass. Scr. Mater. 2010, 63, 879-882.

2. Li F., Liu X. J.; Lu Z. P., Atomic structural evolution during glass formation of a Cu-Zr binary metallic glass. Comput. Mater. Sci. 2014, 85, 147-153.

Reviewer 2 Report

The authors of the paper "Inhomogeneity of free volumes in metallic glasses under tension" describes the results of the metallic glass (MG) tensile test MD simulation. The obtained by the authors non-homogenious increasing of the free volume in the different part of the simulated volume seems to be interesting for the analysis of shear deformation of MG.

The paper may be accepted for the publication after minor revisions:

The level of analysed strains is hardly may achieved in real tensile test due to shear localization immediately after the elastic region. The authors should justify the practical value of the simulation results obtained at high strains such as a chosen strain rate value 108 s-1 which is in 12 order larger than usual strain rates at real deformation.         

Author Response

Reviewer 2

The authors of the paper "Inhomogeneity of free volumes in metallic glasses under tension" describes the results of the metallic glass (MG) tensile test MD simulation. The obtained by the authors non-homogenious increasing of the free volume in the different part of the simulated volume seems to be interesting for the analysis of shear deformation of MG. The paper may be accepted for the publication after minor revisions.

Reply: Thanks the referee very much for his/her positive comments.

1) The level of analysed strains is hardly may achieved in real tensile test due to shear localization immediately after the elastic region. The authors should justify the practical value of the simulation results obtained at high strains such as a chosen strain rate value 108 s-1 which is in 12 order larger than usual strain rates at real deformation.

Reply: I agree with the reviewer that the strain rate used in MD simulation is much larger than that in a real experiment. This is due to the fact the current computing capability cannot support the MD simulation on a metallic glass (MG) model using a small strain rate comparable to that in the real tensile test. I suppose that the simulation time will last for billions of years if we want to do so. A strain rate value 108 s-1 is usually adopted in many MD simulation articles studying the mechanical properties of MGs, such as:

1. Li Q. K., Atomic scale characterization of shear bands in an amorphous metal. Appl. Phys. Lett. 2006, 88, 241903;

2. Feng S. D.; Qi L.; Li G.; Zhao W.; Liu R. P., Effects of pre-introduced shear origin zones on mechanical property of ZrCu metallic glass. J. Non-Cryst. Solids, 2013. 373, 1-4;

3. Nicholas P. Bailey,; Jakob Schiøtz,; and Karsten W. Jacobsen, Atomistic simulation study of the shear-band deformation mechanism in Mg-Cu metallic glasses. Phys. Rev. B, 2006, 73, 064108.

Although compared with the real experiment, there are some problems when we performing a tension test upon a MG model in MD simulation, such as the relatively fast strain rate, we can validate our MD simulation using free volume data to discuss the mechanical behavior in MGs, because: 1) since our model is smaller than the thin films, and even much smaller than BMGs, it is reasonable that our model can exhibit the “best plasticity”, the thin films have “good plasticity”, while BMGs usually have “poor plasticity”. This is quite consistent with the size effect that I have discussed a lot to reply the reviewer 1. In brief, the dependence of plasticity on size in MGs is an intrinsic effect which can be validated by the MD simulation, confirming the reliability of our work; 2) it is found that free volumes in our model get saturated from the strain about 10%, in accord with previous work that there is a critical fraction about 2.4% for reduced free volume from the onset of yielding in both brittle and plastic MGs. This also indicates the reliability of analysis of free volumes in our model. Therefore, I suppose that the “Inhomogeneity of free volumes in metallic glasses under tension” may be an intrinsic feature in our model, thin films, or BMGs, although it is still difficult to perform the in-situ measurement of free volumes in thin films, or BMGs which are under stress.

In addition, based on our work studying the “Inhomogeneity of free volumes in metallic glasses under tension”, in the future, we can further study the change of free volumes in some local regions especially in or close to the places where shear initiation, development, and transformation may occur. This may be helpful to reveal the structural origin of good plasticity in MGs, because it has been suggested that the free volume evolution in the localized shear zones may explain the brittleness or plasticity in MGs. Anyway, in this work, it is still difficult to discuss the plasticity in MGs, whereas we are expecting to address this issue.